# Impact of the COVID-19 Pandemic on Epidemiology of Antibiotic Resistance in an Intensive Care Unit (ICU): The Experience of a North-West Italian Center

**DOI:** 10.3390/antibiotics12081278

**Published:** 2023-08-03

**Authors:** Andrea Parisini, Silvia Boni, Elisabetta Blasi Vacca, Nicoletta Bobbio, Filippo Del Puente, Marcello Feasi, Roberta Prinapori, Marco Lattuada, Marina Sartini, Maria Luisa Cristina, David Usiglio, Emanuele Pontali

**Affiliations:** 1Department of Infectious Diseases, Galliera Hospital, Mura delle Cappuccine 14, 16128 Genoa, Italy; andrea.parisini@galliera.it (A.P.); silvia.boni@galliera.it (S.B.); elisabetta.blasi@galliera.it (E.B.V.); nicoletta.bobbio@galliera.it (N.B.); filippo.del.puente@galliera.it (F.D.P.); marcello.feasi@galliera.it (M.F.); roberta.prinapori@galliera.it (R.P.); 2Anaesthesia and Intensive Care Unit, E.O. Ospedali Galliera, Mura delle Cappuccine 14, 16128 Genoa, Italy; marco.lattuada@galliera.it; 3Department of Health Sciences, University of Genova, Via Pastore 1, 16132 Genoa, Italy; marina.sartini@galliera.it (M.S.); maria.luisa.cristina@galliera.it (M.L.C.); 4Operating Unit (S.S.D.U.O.) Hospital Hygiene Unit, Galliera Hospital, Mura delle Cappuccine 14, 16128 Genoa, Italy; 5Department of Laboratory and Microbiological Analysis, Galliera Hospital, 16128 Genoa, Italy; david.usiglio@galliera.it

**Keywords:** SARS-CoV-2, intensive care unit (ICU), multidrug-resistant microorganism (MDRO), resistance

## Abstract

The SARS-CoV-2 pandemic caused an increase in intensive care unit (ICU) hospitalizations with a rise in morbidity and mortality; nevertheless, there is still little evidence on the impact of the pandemic on antibiotic resistance in ICUs. This is a retrospective, monocentric epidemiological study. The aim of the study was to describe and analyze the impact of the SARS-CoV-2 pandemic on ICU bacterial resistance patterns. All bacteria isolated from all patients admitted to the E.O. Galliera ICU from January 2018 to December 2022 were included. Antibiotic resistance (AR) profiles were evaluated. A total of 1021 microorganisms were identified, of which 221 (12.47%) had a resistance pattern (resistant organisms; ROs). In this time, there were 1679 patients with a total of 12,030 hospitalization days. The majority of microorganisms were Gram-negative (79.66% in 2018, 77.29% in 2019, 61.83% in 2020, 62.56% in 2021, and 60.75% in 2022), but an increase in Gram-positive microorganisms was observed (20.34 to 39.25% between 2018 and 2022). The prevalence of AR was 19.44% in 2018, 11.54% in 2019, 38.04% in 2020, 34.15% in 2021, and 39.29% in 2022 for Gram-positive microorganisms and 19.86% in 2018, 13.56% in 2019, 18.12% in 2020, 12.41% in 2021, and 12.31% in 2012 for Gram-negative microorganisms. The incidence of ROs showed a COVID-19-related increase in 2020–2021, followed by a lowering trend since 2021, and a new increase in 2022. Possible explanations are antibiotic overtreatment and a decrease in containment measures. An interesting finding was the cumulative lowering trend of carbapenem-resistant *K. pneumoniae* and *P. aeruginosa*, probably due to different patient features.

## 1. Introduction

Since 2020, the SARS-CoV-2 pandemic has resulted in a high number of COVID-19 cases with severe respiratory failure who are frequently hospitalized in intensive care units (ICUs) with a well-documented concurrent rise in morbidity and mortality among them [1]. Nevertheless, there is still limited evidence on the real impact that this pandemic has had on the epidemiology of resistant microorganisms (ROs) in this setting [2]. Some retrospective reports have shown a lower incidence of infections due to ROs during the pandemic, with a less significant impact in COVID-19 wards versus non-COVID-19 ones [3]. In particular, in 2021, Grasselli et al. reported bacterial infection incidence at 44.7/1000 ICU days with 35% RO infections among ICU COVID-19 patients [4]. A 2020 case series by Cusumano et al. highlighted the pivotal role of secondary bacterial infections on mortality in COVID-19 patients [5]. A British report showed that bacterial coinfections at admission were detected in 14% of ICU patients, with 85% of them already receiving antibiotics at that stage. This same paper showed that 54% of patients experienced at least one ICU-acquired infection [6]. Langford and colleagues commented that the COVID-19 pandemic may have hastened the emergence, transmission, and diffusion of antimicrobial resistance, particularly for Gram-negative organisms, in hospital settings [7]. However, the same authors reported considerable heterogeneity in both the antimicrobial resistance metrics used and the rate of resistance reported across studies [7]. In addition, several reports have highlighted the changing epidemiology of ROs in ICUs in Europe during different waves of the COVID-19 pandemic and, in particular, the significant increase in multidrug-resistant *A. baumannii* in Italian ICUs [8,9,10,11]. 

The aim of our study was to describe and analyze the impact of the SARS-CoV-2 pandemic on the resistance patterns of ICU microorganisms.

## 2. Materials and Methods

This is a single-center, retrospective study conducted in a general hospital of national relevance that is highly specialized, treatment intensity based, and located in northern Italy. The hospital has a pavilions-based structure of 458 beds, with more than 15,000 routine admissions/year and more than 8600 medical procedures in outpatient and day surgery settings. In particular, the ICU has 8 beds, 6 of which are in an open space area and 2 in single isolated rooms. During the first pandemic wave, the ICU beds were initially increased to 16 for a few months, then remained at 12 for several months and went up and down following different waves of the COVID-19 pandemic [1]. All bacteria isolated from all patients admitted to the E.O. Galliera ICU from January 2018 to December 2022 were included in this study. Their antibiotic resistance profiles were reported by our laboratory following EUCAST guidelines [12]. Samples were collected from the following materials: bronchoaspiration/broncholavage fluid, blood culture and central line blood culture, urine, and others (peritoneal fluid, surgical sites, etc.). Microorganisms were included regardless of their nature (either infection or colonization). For all microorganisms, information regarding date and site of isolation and the presence or absence of antimicrobial resistance was collected.

Data were anonymously collected for all patients admitted in ICU from 1 January 2018 to 31 December 2022. Multiple isolations of the same pathogen with the same resistance pattern and site within 14 days were considered as a single sample for each patient. 

The study was conducted according to the guidelines of the Declaration of Helsinki. Ethical review and approval were waived for this study because all analyzed data were extracted anonymously by the laboratory and the study did not involve humans or animals. 

Statistical Analysis

Data were collected in an Excel database. A descriptive analysis was first performed. The chi-squared test was used to assess independence between variables. A *p*-value of less than 0.05 was considered statistically significant. All statistical analyses were performed using the Stata/SE 14.2 software (StataCorp LP, College Station, TX, USA).

## 3. Results

During the study period (2018–2022), a total of 1021 bacteria were detected in ICU, of which 220 (21.55%) had some resistance pattern (resistant organisms; ROs). In the same period, 1679 patients spent at least one day in the ICU with a total of 12,029 ICU hospitalization days. The abovementioned microorganisms were isolated from 791 patients and were included in the study. Some patients had both ROs and non-ROs during the same stay. Their mean age was 68.33 ± 13.61 years (median 71), and they were mainly males (274; 60.22%) (Table 1).

Most of the isolated microorganisms were Gram-negative (67.68%), of which Enterobacterales represented more than two thirds (70.33%) and nonfermenting ones represented 29.67%. The most frequently detected Gram-positive microorganisms were coagulase-negative Staphylococci (CoNS) (44.24%), Enterococci (23.64%), *Staphylococcus aureus* (17.27%), and Streptococci (14.85%). During the study years, a progressive increase in the proportion of Gram-positive microorganisms occurred, with a statistically significant increase between 2019 and 2020 (*p* < 0.001) (Figure 1). 

Microorganisms were isolated from different biological materials; their proportion per cultured material is reported in Table 2.

Regarding ROs, their absolute number increased from 35 in 2018 to 62 in 2020 and later swayed between 45 and 49. Their prevalence was between 13.10 and 25.73% of all isolated microorganisms (Table 3). Their prevalence changed over the study years, but these changes were not significant.

The incidence of ROs and non-ROs changed over the study period, but this change was not statistically significant (Table 4).

The proportion of ROs increased from 2020, especially for Gram-positive microorganisms. In contrast, Gram-negative microorganisms increased from 13.56 up to 18.12% in 2020, but a new slowly decreasing trend to pre-pandemic levels was observed in 2021–2022 (Figure 2).

The most frequently observed drug resistance patterns among ROs during the study period were for oxacillin at 44.80%, carbapenems at 33.93%, and extended-spectrum beta-lactamase (ESBL) at 16.29%.

### 3.1. Oxacillin Resistance

#### 3.1.1. *S. aureus*

Few strains of *S. aureus* were detected during the study years. Among them, the prevalence of oxacillin resistance (MRSA) fluctuated from an initial 57.14% (4/7) in 2018, down to 30% (3/10) in 2019, stable in 2020 (35.71%; 5/14) and 2021 (27.7%; 3/11), and then going up to 60% (9/15) in 2022 (Figure 3). 

#### 3.1.2. Coagulase-Negative Staphylococci (CoNS)

As expected, more strains of CoNS than *S. aureus* were isolated. The prevalence of oxacillin resistance among CoNS (MR-CoNS) was initially low at 22.22% (2/9) in 2018 and 20% in (3/15) in 2019. In 2020, there was an increase in the proportion of resistant strains, which remained relatively stable in the following two years (2020: 59.52% (25/42), 2021: 60.98% (25/41), and 2022: 48.72% (19/39)).

A temporal correlation between an increase in oxacillin resistance among CoNS and the SARS-CoV-2 pandemic was observed, with a statistically significant increase in 2020 (*p* < 0.01), a further increase in 2021, and then a decrease in 2022 (Figure 3). 

### 3.2. Carbapenem Resistance (CR)

#### 3.2.1. Enterobacterales

The proportion of carbapenem-resistant Enterobacterales (CRE) dropped from 17.82% in 2019 to 5.33% in 2022, showing a nonsignificant lowering trend (R^2^ = 0.82) over the years (Figure 4). 

#### 3.2.2. Nonfermenting Gram-Negative Microorganisms 

In this group, *P. aeruginosa* was the most frequently identified microorganism with carbapenem resistance (20 strains in 5 years vs. 4 *A. baumannii*). The proportion of resistant strains went up and down during the study years. There was a drop in 2019 (8.33%) from the previous year (15.79%), then a nonstatistically significant increase was observed in 2021 (25% of resistant isolates), followed by a new decrease to 10.71% in 2022 (Figure 5).

#### 3.2.3. *K. pneumoniae*

The trend in carbapenem resistance of *K. pneumoniae* prevalence was as follows: 15/29 (51.72%) in 2018, 5/19 (26.32%) in 2019, 7/20 (35%) in 2020, 0/9 (0%) in 2021, and 2/13 (15.38%) in 2022. 

### 3.3. Extended-Spectrum Beta-Lactamase (ESBL)

#### Enterobacterales

Resistant isolates in this group were rare and relatively stable over the study years: 6/101 (5.94%) in 2018, 10/124 (8.06%) in 2019, 7/94 (7.45%) in 2020, 5/92 (5.43%) in 2021, and 8/75 (10.67%) in 2022 (Figure 4).

### 3.4. Incidence Rates 

During the study years, there was a decrease in detection of *E. coli* ESBL strains, which went from 2.681/1000 days of hospitalization in ICU in 2019 to 0.888/1000 ICU days. A similar trend was observed for CR *P. aeruginosa*, which, after an initial rise in 2019–2020 (2.649/1000 ICU days), fell to 1.332/1000 ICU days in 2022. However, both trends were not statistically significant. 

In contrast, for *K. pneumoniae* ESBL, there was an incidence increase up to 2.664/1000 ICU days in 2022 after years of stable incidence.

The highest incidence values were observed for OXA-R *S. epidermidis*, which was 7.946/1000 ICU days in 2020, 8.037/1000 ICU days in 2021, and 6.661/1000 ICU days in 2022 (Figure 6).

## 4. Discussion

The SARS-CoV-2 pandemic had a significant impact on our hospital (as it did on others) in terms of organization, change in infection control measures (greatest attention to respiratory transmission), and type of patients in ICU (less surgical patients and more respiratory ones) but with proportions and categories of patients continuously changing in the various phases of the pandemic. This can be expected to have had an impact on the circulation of microorganisms (resistant or not) too. Although our study has some significant limitations (monocentric, small size, observational, and retrospective), a few findings merit careful consideration. Some important ROs in ICU after a first increase in incidence rates during the early COVID-19 pandemic years (2020–2021) showed a decreasing trend in 2021–2022. The observed rates reached even lower levels than the pre-pandemic period (2018–2019) for *ESBL E. coli*, CR *K. pneumoniae*, and CR *P. aeruginosa*. This would suggest that the changed patient characteristics at ICU admission (younger age, less comorbidity, less surgical patients, and less risk factors for ROs carriage) since the beginning of the SARS-CoV-2 pandemic may have impacted the observed resistance patterns. Different trends were observed for strains such as Gram-negative ESBL and MRSA, which after an initial lowering trend, saw an increase in 2020–2021 and 2022, possibly related to decreased infection control activities, reduced adherence to blood culture and central line bundles, and antibiotic overtreatment in COVID-19 patients. The increase in MRCoNS in 2020 and 2021, especially in CVC blood cultures, with a new decrease in 2022 is probably related to reduced infection control policies, while the CVC management bundle observation is probably due to high staff turnover and the presence of not strictly qualified employees in ICU settings in the first pandemic phases. These observations can be in part confirmed by the new decreasing trends observed at the end of 2021 and 2022 after new personnel training and CVC bundle reinforcement were carried out at the end of 2021. 

One meta-analysis by Kariyawasam et al. in 2022 reported a global rise in antimicrobial resistance (AMR) in COVID-19 patient with MRSA as a leading pathogen in this subset, even though the increase was not statistically significant [13]. Another similar paper in the same year by Langford et al. confirmed that the COVID-19 pandemic was not associated with a change in the incidence or proportion of MRSA [14]. In a 2022 systematic review, Abubakar et al. reported that in more than 50% of the examined papers, an increase in the prevalence/incidence of MRSA infections/colonizations occurred during the COVID-19 pandemic and that this increase ranged from 4.6 to 200.0% [15]. Another retrospective analysis by Segala et al. evaluated bloodstream infection incidence in 2018–2021, i.e., in pre-pandemic and pandemic periods. They confirmed a higher risk of blood stream infection (BSI) occurrence among SARS-CoV-2 positive patients with a more pronounced risk in ICU. *S. aureus* was the second pathogen in frequency and with a MRSA prevalence of more than 40% [16]. It is not clear why this phenomenon occurred in all such settings, i.e., if it was due to increased steroid use, increase in MRSA pneumonia concomitant or subsequent to COVID-19 pneumonia in ICU, or other causes. We actually observed a significantly different pattern with relatively low incidence of MRSA isolation, while detection of CoNS was much more frequent. A first consideration concerning our observation is that this trend is probably related to overcrowding in ICU, massive staff turnover during this period, and the presence of not previously properly trained healthcare workers with consequent insufficient hand hygiene and adherence to central venous catheter (CVC) management bundles in the first SARS-CoV-2 waves.

In a retrospective, cohort study by Punjabi et al. on MRSA prevalence in respiratory cultures of COVID-19 patients with severe pneumonia and critical illness, MRSA coinfection was correlated with prolonged hospital stay, implying that it is more likely to be a hospital-acquired or ventilator-associated complication than a community-acquired coinfection [17]. The main difference between this latter paper and our findings is that the large majority of MRSA isolates at our ICU were obtained from bronchoaspirate (BRA) in 2018 and 2022 and blood cultures in 2019 and 2022. This is in contrast to 2020–2021, where the isolates were all from CVC blood cultures. Due to the increase in CVC-related isolations in our ICU, on-the-job training on CVC management bundles and its strict monitoring was implemented since autumn 2021. A finding in favor of the effectiveness of this intervention is the proportional rise of MRCoNS, especially in CVC blood cultures in the first years of the pandemic in 2020–2021, followed by a progressive lowering curve at the end of 2021 and then 2022. However, we need to underscore that our study was only epidemiological; thus, the real clinical impact and role of MRCoNS CVC blood culture positivity has to be carefully considered. Unfortunately, the sample size was too small for further analysis, and no significant trend was identified in our study.

Regarding Gram-negative Ros, several published reports have shown a variable epidemiology with different emerging pathogens in different settings [8,9,10,11]. In one retrospective study, the colonization by CRE increased from 6.7 to 50% from 2019 to 2020 [18]. Another study on the Italian experience by Temperoni et al. [19] reported a high prevalence of *A. baumannii* carbapenem-resistant colonization with an infection rate of 75% (12/16). As expected, 100% of isolates of *P. aeruginosa* in that paper were multidrug-resistant (MDR), including four carbapenem-resistant (CR) strains. Similarly, 50% of *K. pneumoniae* isolates were MDR, while only 31% of isolates of *E. coli* resulted in MDR [8,11]. Similarly, in a recent retrospective study, Gajic et al. observed a high rate of *K. pneumoniae* and *A. baumannii*, which together accounted for 47 and 75.8% of the most common types of bloodstream and respiratory tract infections, respectively [20]. Another Italian paper by Tiri et al. underscored a tight connection between COVID-19 and an increase in CRE colonization and infection in ICU. In this latter study, the authors analyzed bimonthly data collected from January–February 2019 to May–June 2020 concerning the number of patients admitted to the ICU and the number of rectal carriers of CRE. The incidence of CRE acquisition ranged from 5% on average in 2019 to 50% during the pandemic (March–April 2020). Later, in May–June 2020, when the ICU was again COVID-19 free, a new drastic decreasing trend from 50 to 0% was observed [21]. The most striking difference with our findings is the low level of resistant *A. baumannii* strains detected in our cohort. This is probably and partially explained by a historically very low environmental prevalence and circulation of *Acinetobacter* in our hospital and ICU in particular; furthermore, every single case (colonization or infection) is always promptly and strictly isolated and managed with dedicated staff as soon as it is detected. 

Recent literature has reported variable results for ESBL-producing Enterobacterales too. Lemenand et al. investigated the impact of the COVID-19 pandemic on the epidemiology of *E. coli* ESBL+ in France. After analyzing clinical samples from primary care patients and nursing home residents collected between January 2019 and December 2020, they found an intriguing reduction in the percentage of *E. coli* ESBL+ isolates. This study reported several limitations, including the lack of contextual information to determine the relative contributions of factors such as decreased healthcare utilization, reduction in antibiotic dispensing, and the fact that the study only explored the proportion of *E. coli* ESBL+ rates and no other microorganisms. Nevertheless, it is interesting to observe such an apparent favorable impact of the national COVID-19 pandemic response on the *E. coli* ESBL+ epidemiology in primary care settings and nursing homes in France [22]. In our study, isolation of ESBL pathogens was stable. A possible explanation can be found in the patient features at admission in ICU during the pandemic years. COVID-19 patients, especially in the first phases of the pandemic, were generally younger with less comorbidity and less classic risk factors for ESBL carriage at hospital admission. 

Concerning CRE circulation, our data, in contrast to many other studies, showed that isolation of these microorganisms dropped during the study period, meaning that active antimicrobial stewardship initiatives and isolation bundles were probably ongoing despite the logistics and staff difficulties occurring during the pandemic.

## 5. Conclusions

In our study, we observed peculiar trends in the incidence of both Gram+ and Gram− ROs. In comparison with other reports, we observed different epidemiological patterns. Even other reports have not been unanimous in their findings. Further analysis of globally collected data will be necessary to understand if general trends in the isolation of ROs in ICU had similar patterns or if each hospital/region/country had peculiarities due to the local pre-COVID-19 epidemiology, local infection control practices, and specific interventions implemented.

In our setting, as soon as COVID-19 pressure on ICU wards decreased, more antimicrobial stewardship activities were again in place, including regular and more frequent infectious disease (ID) specialist consultations, more frequent de-escalation and shortening of antibiotic treatment, and renewed higher efforts in implementing the infection control strategy. The combination of these efforts, along with patient features at hospital presentation, particularly in the first phase of the pandemic, were probably the main drivers towards incidence reduction for most ROs in our ICU setting. This is particularly striking regarding the very low occurrence of CR-*A. baumannii* and CRE. 

Further larger and more detailed investigations are indeed needed to better assess the real impact of the SARS-CoV-2 pandemic on the incidence of ROs and consequently infections and related mortality in ICU.

## Figures and Tables

**Figure 1 antibiotics-12-01278-f001:**
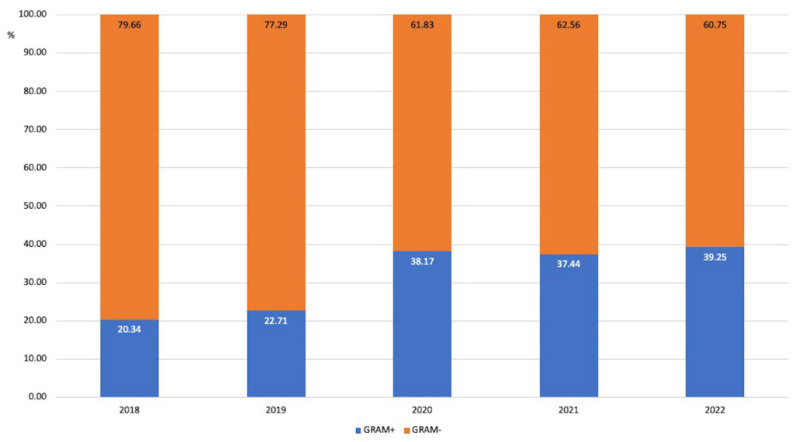
Proportion of Gram-positive and Gram-negative microorganisms among the isolated microorganisms during the observation time.

**Figure 2 antibiotics-12-01278-f002:**
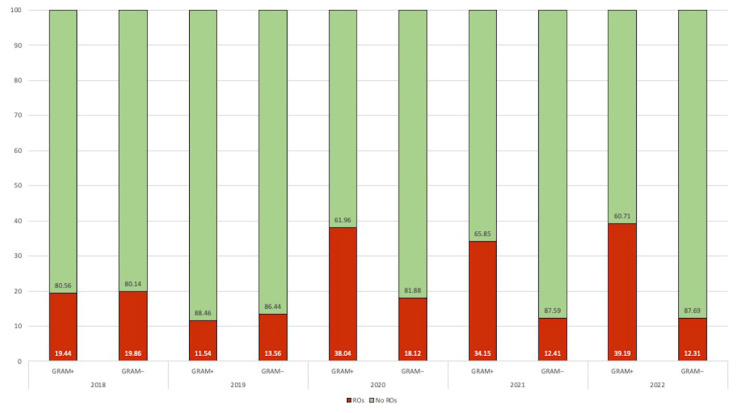
Distribution of resistant organisms (ROs) and sensitive Gram-positive and Gram-negative pathogens according to the years.

**Figure 3 antibiotics-12-01278-f003:**
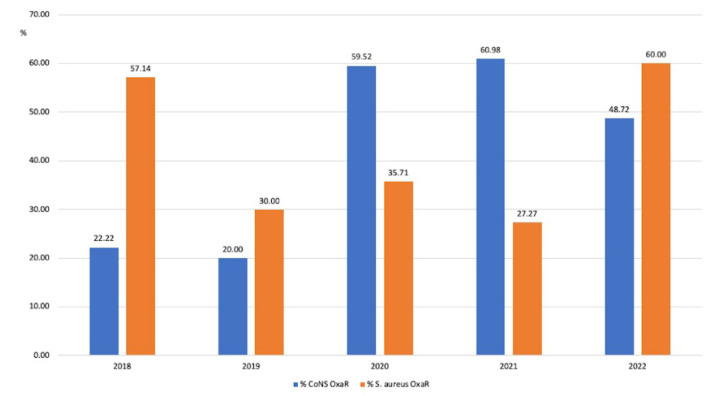
Changes in the prevalence of oxacillin resistance (%) among *S. aureus* and coagulase-negative Staphylococci (CoNS) isolates.

**Figure 4 antibiotics-12-01278-f004:**
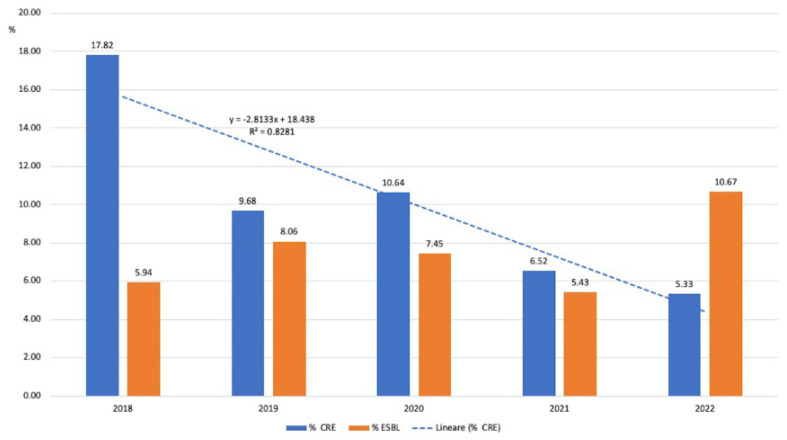
Evolution of CR and ESBL (%) in Enterobacterales.

**Figure 5 antibiotics-12-01278-f005:**
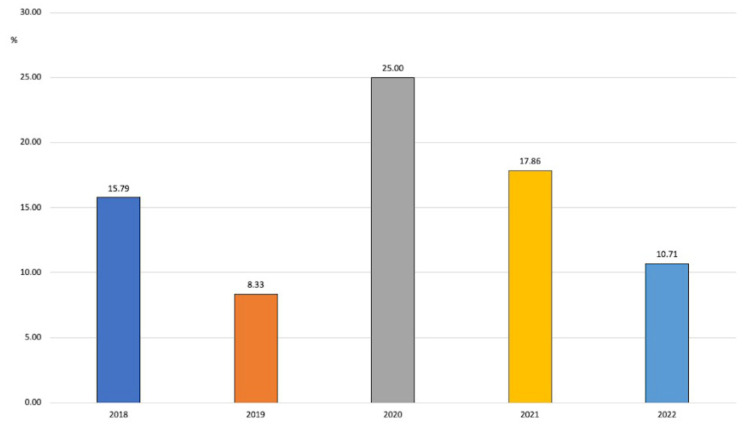
Evolution of carbapenem resistance in *P. aeruginosa* over the years.

**Figure 6 antibiotics-12-01278-f006:**
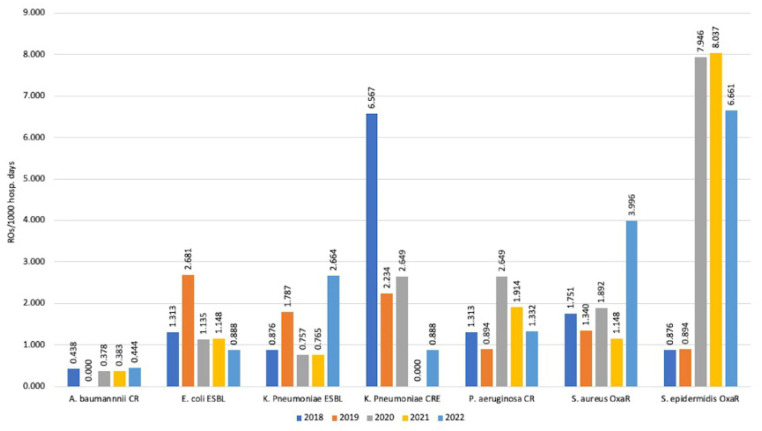
Incidence rate for the main resistant pathogens observed.

**Table 1 antibiotics-12-01278-t001:** Characteristics of patients with at least one microorganism isolation.

	All (Number of Patients = 791)	Non-ROs (Number of Patients = 730)	ROs (Number of Patients = 180)
Age (years)			
Mean ± SD	68.63 ± 13.61	68.64 ± 13.94	68.56 ± 12.06
Median	71	71	69
Gender			
Female	181 (39.78%)	158 (42.36%)	23 (28.05%)
Male	274 (60.22%)	215 (57.64%)	59 (71.95%)
Number of isolates	1021	801	220

**Table 2 antibiotics-12-01278-t002:** Proportion of isolated microorganisms per cultured material.

Year	BRA/BAL	Blood Culture	CVC- Blood Culture	Urine	Others
2018	105 (62.88%)	14 (8.38%)	7 (4.19%)	28 (16.77%)	13 (7.78%)
2019	121 (57.90%)	17 (8.13%)	7 (3.35%)	43 (20.57%)	21 (10.05%)
2020	152 (66.08%)	14 (6.09%)	16 (6.96%)	37 (16.09%)	11 (4.78%)
2021	126 (58.60%)	19 (8.84%)	15 (6.98%)	36 (16.74%)	19 (8.84%)
2022	97 (48.50%)	28 (14.00%)	8 (4.00%)	38 (19.00%)	29 (14.50%)

BRA: bronchoaspirate. BAL: broncholavage. CVC: central venous catheter. Others: this group includes peritoneal fluid and surgical site infections.

**Table 3 antibiotics-12-01278-t003:** Distribution between non-ROs and ROs during the observation time.

	Non-ROs (%)	ROs (%)	Patients (N)
2018	142 (80.23)	35 (19.77)	375
2019	199 (86.90)	30 (13.10)	371
2020	179 (74.27)	62 (25.73)	337
2021	174 (79.45)	45 (20.55)	309
2022	165 (77.10)	49 (22.90)	287

**Table 4 antibiotics-12-01278-t004:** Incidence of non-ROs and ROs during the study period.

	Non-ROs/1000 pt Days	ROs/1000 pt Days	Hospitalization Days
2018	62.172	15.324	2284
2019	88.919	13.405	2238
2020	67.726	23.458	2643
2021	66.590	17.222	2613
2022	73.268	21.758	2252

## Data Availability

The data presented in this study are available on request from the corresponding author. The data are not publicly available due to internal regulations.

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
