# Peer review of "Impact of the COVID-19 Pandemic on Epidemiology of Antibiotic Resistance in an Intensive Care Unit (ICU): The Experience of a North-West Italian Center"

_antibiotics, 2023, doi:10.3390/antibiotics12081278_

Round 1

Reviewer 1 Report

The article's structure was well-organized, making it easy to follow along. It began with a compelling introduction that immediately grabbed my attention, posing questions and introducing a fascinating topic.

As a minor, I appreciated their ability to make complex concepts accessible and relatable to readers of all ages.

Please replace the term Enterobacteriaceae with Enterobacterales.

Please reformulate paragraph Line 51.

As a minor, I often find that visual aids enhance my understanding and make the content more engaging. Integrating relevant images, graphs, or diagrams would have further elevated the article's appeal. Please reformulate figure number 5.

While there was room for improvement in terms of visual aids, the article's overall impact on me as a minor reader was undeniably positive. I eagerly look forward to discovering more thought-provoking works by this talented author in the future.

The article's structure was well-organized, making it easy to follow along. It began with a compelling introduction that immediately grabbed my attention, posing questions and introducing a fascinating topic.

As a minor, I appreciated their ability to make complex concepts accessible and relatable to readers of all ages.

Please replace the term Enterobacteriaceae with Enterobacterales.

Please reformulate paragraph Line 51.

As a minor, I often find that visual aids enhance my understanding and make the content more engaging. Integrating relevant images, graphs, or diagrams would have further elevated the article's appeal. Please reformulate figure number 5.

While there was room for improvement in terms of visual aids, the article's overall impact on me as a minor reader was undeniably positive. I eagerly look forward to discovering more thought-provoking works by this talented author in the future.

Author Response

The article's structure was well-organized, making it easy to follow along. It began with a compelling introduction that immediately grabbed my attention, posing questions and introducing a fascinating topic.

Thank you for your positive feedback and for your suggestions to improve the paper

As a minor, I appreciated their ability to make complex concepts accessible and relatable to readers of all ages.

Please replace the term Enterobacteriaceae with Enterobacterales.

Replaced as suggested

Please reformulate paragraph Line 51.

Sentence was modified

As a minor, I often find that visual aids enhance my understanding and make the content more engaging. Integrating relevant images, graphs, or diagrams would have further elevated the article's appeal. Please reformulate figure number 5.

Thanks for the suggestion. However, we checked the figure and we decided to keep it as it was since each color represents a different year and we think it is sufficiently clear.

While there was room for improvement in terms of visual aids, the article's overall impact on me as a minor reader was undeniably positive. I eagerly look forward to discovering more thought-provoking works by this talented author in the future.

In addition a careful revision of Tables with changes in reported figures occurred

Reviewer 2 Report

This is interesting epidemiological study; although with limitations (small-size, monocentric, retrospective), it deserves to be published. However, some points should be revised. The major ones are listed bellow, while the minor points are provided directly within the revised manuscript (provided as attach file).

1. The information concerning the approvement of the study by the appropriate Ethics Committee has to be provided. It could be noted within the Material and Method Section.

2. The Conclusion Section should be revised. In the submitted manuscript version, authors discussed previously published researches within the Conclusion Section; however, that should be avoided.

3. The use of abbreviations should be carefully checked. The authors used some of them without their clear involvement at the place of first mentioning, while the others were redundant, involved almost at the end of the manuscript and not used in the following text.

Author Response

This is interesting epidemiological study; although with limitations (small-size, monocentric, retrospective), it deserves to be published.

Thank you for your positive feedback and for your suggestions to improve the paper

However, some points should be revised. The major ones are listed bellow, while the minor points are provided directly within the revised manuscript (provided as attach file).

  1. The information concerning the approvement of the study by the appropriate Ethics Committee has to be provided. It could be noted within the Material and Method Section.

It has been clarified in the text that for this study approval by the Ethics Committee was not necessary.

  1. The Conclusion Section should be revised. In the submitted manuscript version, authors discussed previously published researches within the Conclusion Section; however, that should be avoided.

The conclusions section has been widely revised

  1. The use of abbreviations should be carefully checked The authors used some of them without their clear involvement at the place of first mentioning, while the others were redundant, involved almost at the end of the manuscript and not used in the following text.

Thanks for the note. We identified a few instances where abbreviations had not been properly mentioned and spelt out at first appearance. We carefully checked all and now the text is correct.

In addition a careful revision of Tables with changes in reported figures occurred

Reviewer 3 Report

In my opinion, this research work is interesting due to the importance of SARS-CoV-2 infection on the one hand and the emergence of resistant bacterial strains on the other.

I believe that, to be able to publish this manuscript, it is necessary to better organize the information and structure of it, in such a way that it is more understandable.

I think the authors could consider the following observations.

The title and footnote to Figure 1 are out of place.

The authors must clearly explain how the percentage of ROs in 2021 is very similar to that in 2018 and how it is explained that said percentage has decreased in 2019.

Clearly explain how the data presented in Table 3 correlates with the data presented in Figure 2, there would appear to be an inconsistency.

There are very important differences between S. aureus and Coagulase negative Staphylococci (CONs), which I think the authors should highlight.

Resistant S. aureus, especially MRSA, has been associated with increased pathogenicity and severe disease.

On the other hand, CONs are usually related to low pathogenic species, and represent nosocomial pathogens, with S. epidermidis and S. haemolyticus being the most significant species.

Figure 3 shows that between 2019 and 2021 resistant S. aureus isolates (more virulent) decreased and resistant CONs increased. Only in 2022 did an increase in resistant S. aureus isolates begin to occur, being less than CONs. I think that this data should be highlighted more by the authors in their conclusions and discussions.

I also suggest that the authors emphasize that as shown in Figure 4 In the period between 2019 to 2021 EBSL showed a downward trend.

In section 2.3.1. that addresses the ESBL results at the end mentioned in Figure 3, but the latter deals with the results between S. aureus and CONs.

I consider that the reference to Figure 5 should not go at the end of the title of point 2.2.2. but at the end of the text of this. The same situation for points 2.3.1. and 2.4., 2.2.1. and 2.1.

Author Response

In my opinion, this research work is interesting due to the importance of SARS-CoV-2 infection on the one hand and the emergence of resistant bacterial strains on the other.

Thank you for your positive feedback and for your suggestions to improve the paper

I believe that, to be able to publish this manuscript, it is necessary to better organize the information and structure of it, in such a way that it is more understandable.

Thanks for your suggestions. We tried to better organize the paper based on your suggestions and those of the other referees.

I think the authors could consider the following observations.

The title and footnote to Figure 1 are out of place.

We modified them

The authors must clearly explain how the percentage of ROs in 2021 is very similar to that in 2018 and how it is explained that said percentage has decreased in 2019.

Several sentences have been added in the discussion

Clearly explain how the data presented in Table 3 correlates with the data presented in Figure 2, there would appear to be an inconsistency.

Thanks for pointing out this inconsistency. Actually, there was a mistake that has now been amended.

There are very important differences between S. aureus and Coagulase negative Staphylococci (CONs), which I think the authors should highlight.

Resistant S. aureus, especially MRSA, has been associated with increased pathogenicity and severe disease.

On the other hand, CONs are usually related to low pathogenic species, and represent nosocomial pathogens, with S. epidermidis and S. haemolyticus being the most significant species.

Several sentences have been added in the discussion

Figure 3 shows that between 2019 and 2021 resistant S. aureus isolates (more virulent) decreased and resistant CONs increased. Only in 2022 did an increase in resistant S. aureus isolates begin to occur, being less than CONs. I think that this data should be highlighted more by the authors in their conclusions and discussions.

Several sentences have been added in the discussion

I also suggest that the authors emphasize that as shown in Figure 4 In the period between 2019 to 2021 EBSL showed a downward trend.

Several sentences have been added in the discussion

In section 2.3.1. that addresses the ESBL results at the end mentioned in Figure 3, but the latter deals with the results between S. aureus and CONs.

This has been modified to be correct now

I consider that the reference to Figure 5 should not go at the end of the title of point 2.2.2. but at the end of the text of this. The same situation for points 2.3.1. and 2.4., 2.2.1. and 2.1.

We corrected all them

In addition a careful revision of Tables with changes in reported figures occurred

Reviewer 4 Report

Recommendations are in the attached file. 

I congratulate the authors on an interesting study.
Special attention must be paid to the corrections of figures. They should be presented more professionally to get them published. They cannot look like drafts. And after the modifications, the work can be published.
Abstract
1. 21: The naming of a type of study appears as a fragment of a sentence in the context.
2. 30: resistant organisms indicated as ‘‘Ros“, not ROs. Please, unify.
3. The aim of the study could be indicated before the description of methods.
Introduction:
4. 39, 51: COVID-19, in the whole text - only – COVID.
5. 59: ,,The aim....“ or ,,The study was aimed to...“
Results:
6. The description of Figure 1 is under the tabale 1 and the figure is above table 1. Please correct.
7. Table 1. Instead of ,,No patents“ – n = 791 and etc.
8. Figure 1,2,3,4,5 Please make the text larger in the figure (x,y axes also legends). The design of the figures looks like a draft.
9. 98: Enterobacteriaceae, – should be Italic.
10. 100, 132, 138: Coagulase–negative Staphylococci (CONs) – in small caps.
11. 103: (...) instead of [...] and in the rest of the text when there are links to pictures.
12. Figure 5. I suggest to use the same color of columns because they indicate the same starin of bacteria.
13. 164-167: misleading information. That could be in another chapter.
14. 171-175: If the results are not statistically significant, they should not be should not be discussed as it confuses the reader. The statement is told, then refuted by saying it is statistically insignificant.
Disscusion:
15. Need more discussion.
16. 181-185: introduction, not a discussion. The first paragraph is more suitable for introduction, not for discussion if authors only describe the results from other publications, not the results from the current work. Generally speaking, the discussion should be structured as follows: Authors present their obtained results and how they relate to the results of other researchers. Then follow the explanation of the differences between the authors' results and those of other researchers.
References:
Please include more references.

Author Response

Recommendations are in the attached file. 

I congratulate the authors on an interesting study.

Thank you for your positive feedback and for your suggestions to improve the paper

Special attention must be paid to the corrections of figures. They should be presented more professionally to get them published. They cannot look like drafts. And after the modifications, the work can be published.

Graphical quality of Figures has been improved for all them.

Abstract
1. 21: The naming of a type of study appears as a fragment of a sentence in the context.

Corrected

  1. 30: resistant organisms indicated as ‘‘Ros“, not ROs. Please, unify.

Unified.

  1. The aim of the study could be indicated before the description of methods.

Done

Introduction:
4. 39, 51: COVID-19, in the whole text - only – COVID.

We revised the text and made the use of the acronym uniform throughout the text

  1. 59: ,,The aim....“ or ,,The study was aimed to...“

Changed as suggested

Results:
6. The description of Figure 1 is under the tabale 1 and the figure is above table 1. Please correct.

Changed as per suggestion

  1. Table 1. Instead of ,,No patents“ – n = 791 and etc. NO

We changed to Nr patients. We prefer this version for better clarity

  1. Figure 1,2,3,4,5 Please make the text larger in the figure (x,y axes also legends). The design of the figures looks like a draft.

Graphical quality of Figures has been improved for all them.

  1. 98: Enterobacteriaceae, – should be Italic.

It was changed to Enterobacterales as per rev. 1’s suggestions, thus italics was not anymore necessary

  1. 100, 132, 138: Coagulase–negative Staphylococci (CONs) – in small caps.

Changed throughout the text

  1. 103: (...) instead of [...] and in the rest of the text when there are links to pictures.

(…) were used

  1. Figure 5. I suggest to use the same color of columns because they indicate the same starin of bacteria.

Thanks for the suggestion. However, we checked the figure and we decided to keep it as it was since each color represents a different year and we think it is sufficiently clear.

  1. 164-167: misleading information. That could be in another chapter.

We modified the paragraph

  1. 171-175: If the results are not statistically significant, they should not be should not be discussed as it confuses the reader. The statement is told, then refuted by saying it is statistically insignificant.

We disagree. All information derived from the study is worth discussion; even non-significant results that must be discussed and hypothesis on lack of significance must be included

Disscusion:
15. Need more discussion.

We agree. We expanded discussion

  1. 181-185: introduction, not a discussion. The first paragraph is more suitable for introduction, not for discussion if authors only describe the results from other publications, not the results from the current work. Generally speaking, the discussion should be structured as follows: Authors present their obtained results and how they relate to the results of other researchers. Then follow the explanation of the differences between the authors' results and those of other researchers.

We structured differently our discussion

References:
Please include more references.

Done

In addition a careful revision of Tables with changes in reported figures occurred

Round 2

Reviewer 3 Report

The authors have complied with the suggested observations